# Fast and Accurate Fisher-Guided Quantization via Efficient Kronecker Factor Approximation

## Abstract

Quantization with second-order information has shown strong promise for preserving model quality under aggressive compression. Building on the recent YAQA framework Tseng et al. (2025b), which employs Kronecker-factored approximations of the Hessian via a power-iteration technique, we propose an alternative approach that replaces this step with a more efficient Kronecker decomposition method from Chekalina et al. (2025). This formulation preserves the benefits of second-order curvature-aware quantization while substantially reducing computational cost.

We apply our method to LLaMA-2 7B, LLaMA-3 8B Instruct, Qwen 3 8B Instruct and demonstrate that it achieves the same post-quantization model quality as YAQA, but with significantly faster computational process — the Kronecker factors which provide the required quality was obtained with 10 times fewer tokens and approximately a $10\times$ speedup over the original work.

## 1 Introduction

Large language models (LLMs) have accelerated progress across a wide range of downstream applications. However, their size and computational demands remain prohibitive, making post-training compression a critical research direction.

The standard post-training compression setting assumes that:

1. The model is already trained, and its parameters are at an optimum; therefore, the first-order derivative of the loss is zero and carries no additional information.

2. The second-order derivative characterizes the curvature of the loss surface and highlights the most important directions for compression in parameter space.

3. The goal is to select the most effective compression method from the set of all available approaches.

Since the first-order derivative vanishes at the optimum, effectiveness of the compression must rely on second-order information, i.e., the Hessian of the loss. Specifically, for a layer with weights $\mathbf{W}^\star \in \mathbb{R}^{m \times n}$ for the dataset $D$ Hessian can be defined as:

$$\nabla^2_{\mathbf{W}^\star} L = \mathcal{I}_F(\theta) = \frac{1}{|D|} \sum_{i=1}^{|D|} \text{vec}(\nabla_{\mathbf{W}^\star} \ell) \text{vec}(\nabla_{\mathbf{W}^\star} \ell)^T \in \mathbb{R}^{mn \times mn} \tag{1}$$

Within this framework, we focus on quantization as the compression method. The post-training quantization (PTQ) problem can be formulated as minimizing the second-order Taylor expansion of the loss around the optimum:

$$\arg \min_{\mathbf{W} \in C} \approx \frac{1}{2} (\mathbf{W} - \mathbf{W}^\star)^T (\nabla^2_{\mathbf{W}^\star} L)(\mathbf{W} - \mathbf{W}^\star) \tag{2}$$

where $\mathbf{W}$ denotes the obtained low-precision layer weights after quantization, $\mathbf{W}^\star$ the original high-precision weights, and $C$ the set of possible quantization algorithms.

As shown in Chekalina et al. (2025), we assume that the layer weights follow a multivariate normal (MVN) distribution. Under this assumption, the Fisher information $\mathcal{I}_F$—and consequently the Hessian—can be expressed as a Kronecker product of the inverted row and column covariance matrices, $\mathbf{\Sigma}_{\text{row}}$ and $\mathbf{\Sigma}_{\text{col}}$.

$$\nabla^2_{\mathbf{W}^\star} L \approx \mathcal{I}_F(\theta) = \mathbf{\Sigma}_{\text{col}}^{-1} \otimes \mathbf{\Sigma}_{\text{row}}^{-1} = \boldsymbol{H}_I \otimes \boldsymbol{H}_O \tag{3}$$

We build upon YAQA Tseng et al. (2025b), whose rounding algorithm incorporating second-order information is given by:

$$\mathbf{W} = Q(\mathbf{W}^\star + \mathbf{L_O}^\top \Delta \mathbf{W}\, \mathbf{L_I} + \mathbf{L_O}^\top \Delta \mathbf{W} + \Delta \mathbf{W}\, \mathbf{L_I}) \tag{4}$$

where $\Delta \mathbf{W} = \mathbf{W}^\star - \mathbf{W}$, and $\mathbf{L_O}$ and $\mathbf{L_I}$ are the matrices obtained from the LDL decomposition of $\mathbf{H_O}$ and $\mathbf{H_I}$ from Eq. 3, respectively. As can be seen, in second-order–based quantization the overall procedure naturally decomposes into two parts: (i) the computation of factor matrices that capture second-order information, and (ii) the rounding algorithm, which takes these matrices as parameters.

The core quantization algorithm of YAQA is the QTIP algorithm (Tseng et al. (2025a)), which improves efficiency by transforming model weights to behave like independent Gaussian variables and then applying Gaussian source coding. This transformation not only enables more efficient quantization, but also makes the Gaussian assumption underlying Eq. 3 more justified.

While the method achieves state-of-the-art results, the overall quantization procedure remains time-consuming, primarily due to the second part – the computation of Kronecker factors. Obtaining accurate Kronecker factors of the Fisher information has long been a challenge, as the Hessians of LLM layers are prohibitively large. In YAQA, these factors are estimated using the power iteration method. We propose replacing it with a Lanczos-based method, FastKron, originally introduced in Chekalina et al. (2025). We show theoretically that FastKron converges faster than the power iteration method, and empirically confirm this on modern LLMs, including LLaMA-2 7B, LLaMA-3 8B, and Qwen-3 8B. In the context of second-order PTQ, we do the following:

- We are the first to apply FastKron for efficient computation of exact factors for post-training quantization guided by Kronecker-factored curvature.
- We theoretically show that for LLMs this approach converges faster than the power iteration method proposed in YAQA.
- In modern LLMs, our method empirically achieves a speed increase of about $10\times$ while retaining downstream quality.

Our contribution is a *drop-in* improvement in second-order quantization pipelines: it retains strong quantization algorithms while providing faster factor computation. This approach preserves accuracy and reduces the data- and compute-related burden of curvature estimation, making second-order PTQ more practical at the LLM scale.

## 2 RELATED WORK

Post-training quantization (PTQ) has emerged as a practical approach to reducing the deployment cost of large language models. Existing methods fall broadly into two categories: those that exploit curvature information to guide sensitivity-aware quantization and those that design stronger quantizers through distributional transforms.

**Gaussianization and high-dimensional quantizers.** QuIP# Tseng et al. (2024) applies randomized Hadamard transforms to decorrelate and Gaussianize weight distributions, improving incoherence and enabling more efficient use of lattice and TCQ codebooks. QTIP Tseng et al. (2025a) combines a Gaussianization transform with a bit-shift–based codebook, making the weight distribution more isotropic and better aligned with high-dimensional source coding assumptions. This

allows efficient trellis-coded quantization (TCQ) at scale, significantly improving the rate–distortion trade-off in post-training settings.

**Second-order PTQ.** Curvature-aware PTQ leverages Hessian/Fisher structure to predict sensitivity. HAWQ and HAWQ-V2 Dong et al. (2020) allocate mixed precision by analyzing Hessian spectra, BRECQ Li et al. (2021) reconstructs layer blocks using a second-order error model, and GPTQ Frantar et al. (2022) shows that efficient blockwise approximations of the Fisher or Hessian are sufficient for scaling PTQ to large transformers at 3–4 bits. YAQA Tseng et al. (2025b) provides an adaptive rounding rule that consumes Kronecker-factored layerwise Hessians defined w.r.t. full-model KL divergence—but its dominant cost is computing accurate Kronecker factors.

**Kronecker-factored curvature.** The Hessians of LLM layers are computationally intractable, which has always made obtaining accurate Kronecker factors of the Fisher information challenging. Several approaches, such as K-FAC Martens & Grosse (2020) and FWSVD Hsu et al. (2022b), estimate these factors using diagonal approximations. K-FAC approximates each layer's Fisher matrix as a Kronecker product of two smaller covariance matrices that capture input activations and output gradients, allowing efficient inversion and updates Martens & Grosse (2015). EKFAC George et al. (2018); Bae et al. (2018) refines the K-FAC approximation by re-expressing the Fisher matrix in a Kronecker-factored eigenbasis—diagonalizing each factor and rescaling them using empirical second-order statistics. FWSVD Hsu et al. (2022a) and TFWSVD Hua et al. (2022) adopt a diagonal approximation of the Fisher information for low-rank compression, aligning the factorization objective with parameter importance. GFWSVD Chekalina et al. (2025) extends this approach by exploiting Kronecker-factored Fisher structure and introducing an efficient Lanczos-based factor computation, which we adapt here to produce Kronecker factors directly usable by YAQA/QTIP-style PTQ. Models Eschenhagen et al. (2024) propose a fully optimization-driven approach for estimating Kronecker factors, departing from traditional covariance-based estimators. Instead of computing closed-form second-order statistics, the method treats the factor matrices themselves as learnable parameters and updates them directly using stochastic gradient descent alongside the model weights.

# 3 METHODOLOGY

We quantize LLMs using the YAQA pipeline. To incorporate second-order curvature information from the loss surface, the original work Tseng et al. (2025b) defines the Hessian as:

$$(\nabla^2_{\mathbf{W}^\star} L)_A = \mathbb{E}_{\mathbf{x} \sim \mathcal{D}}\left[\mathbf{x}^T\mathbf{x} \otimes (\nabla_{\mathbf{y}}\ell)^T(\nabla_{\mathbf{y}}\ell)\right], \tag{5}$$

where $\mathbf{x}$ denotes the input activations and $\mathbf{y}$ denotes the corresponding output activations.

Following Loan & Pitsianis (1992), Kronecker factors can be defined as reshaped leading triplets of the SVD of a permuted Fisher information matrix. Based on this formulation, Tseng et al. (2025b) naturally assumes the use of the power iteration method (Golub & Van Loan (2013)) to obtain $\mathbf{H_I}$ and $\mathbf{H_O}$ from Eq. 3 and proposes an algorithm called **Sketch A**. The iterative update at step $i$ is given by:

$$\begin{aligned}
(\mathbf{H}_I)_i &\leftarrow \mathbb{E}_{\mathbf{x} \sim \mathcal{D}}\left[\mathbf{x}^T\mathbf{x}\,(\mathbf{H}_O)_{i-1},\ (\nabla_{\mathbf{y}}\ell)^T(\nabla_{\mathbf{y}}\ell)\right] / \|(\mathbf{H}_O)_{i-1}\|_F^2, \\
(\mathbf{H}_O)_i &\leftarrow \mathbb{E}_{\mathbf{x} \sim \mathcal{D}}\left[(\nabla_{\mathbf{y}}\ell)^T(\nabla_{\mathbf{y}}\ell),\ (\mathbf{H}_I)_{i-1},\ \mathbf{x}^T\mathbf{x}\right] / \|(\mathbf{H}_I)_{i-1}\|_F^2.
\end{aligned} \tag{6}$$

In contrast, for Kronecker factor estimation we introduce **FastKron**, based on the Lanczos algorithm (Lanczos (1950)), originally developed to incorporate second-order information into low-rank pruning. Chekalina et al. (2025) showed that $\mathcal{I}_F$ from Eq. 1 can be rewritten as $\tilde{\mathcal{I}}_F = \frac{1}{|D|}\sum_{i=1}^{|D|}(\nabla_{\mathbf{W}^\star}\ell) \otimes (\nabla_{\mathbf{W}^\star}\ell)^T$.

By exploiting properties of the Kronecker product, this formulation reduces the Lanczos-based SVD of the originally Hessian-sized matrix to matrix multiplications of the same size as a single linear layer — a computation that is tractable on modern GPUs.

The Lanczos-based FastKron algorithm to obtain the Kronecker factors is as follows:

---

**Algorithm 1** FastKron

---

**Require:** List of gradients $\{\nabla_{\mathbf{W}^\star}\ell\}_{i=1}^{|D|}$, $|D|$ – number of batches
1: $\mathcal{I}_F \leftarrow \frac{1}{|D|} \sum_{i=1}^{|D|} \mathrm{vec}(\nabla_{\mathbf{W}^\star}\ell)\mathrm{vec}(\nabla_{\mathbf{W}^\star}\ell)^T$
2: $\tilde{\mathcal{I}}_F = \mathcal{R}\mathcal{I}_F \leftarrow \frac{1}{|D|} \sum_{i=1}^{|D|} (\nabla_{\mathbf{W}^\star}\ell) \otimes (\nabla_{\mathbf{W}^\star}\ell)^T$
3: $(\mathbf{u}, \sigma, \mathbf{v}^\top) \leftarrow$ Leading singular triplet                                         ▷ Truncated SVD
4: $\mathbf{b} \leftarrow \mathbf{u} \cdot \sigma$                                                                         ▷ $\mathbf{b} = \mathrm{vec}(\mathbf{H_I})$
5: $\mathbf{a} \leftarrow \mathbf{v}$                                                                                       ▷ $\mathbf{a} = \mathrm{vec}(\mathbf{H_O})$
6: $\mathbf{H_I} \leftarrow \mathrm{reshape}(\mathbf{b}, (m, m))$
7: $\mathbf{H_O} \leftarrow \mathrm{reshape}(\mathbf{a}, (n, n))$
8: **return** $(\mathbf{H_I}, \mathbf{H_O})$

---

Both SVD-based algorithms — power iteration and Lanczos — have distinct advantages and limitations, and their efficiency varies depending on the properties of the data to which they are applied. A key parameter that influences convergence speed is the spectral gap, i.e., the relative magnitude of the first singular value compared to the second. In the following section, we provide general convergence estimates for each algorithm and analyze the extent to which real-world LLM data (specifically, from LLaMA-2 7B) lie in the regime where one or the other method is more efficient.

**Theorem 3.1.** *Consider a symmetric matrix $\mathbf{A} \in \mathbb{R}^{n \times n}$ with eigenvalues $\lambda_1 \geq \lambda_2 \geq \cdots \geq \lambda_n \geq 0$ and corresponding orthonormal eigenvectors $\mathbf{u}_1, \ldots, \mathbf{u}_n$. The normalized spectral gap is defined as $q = \frac{\lambda_2}{\lambda_1}, \quad 0 < q < 1, q \in (0, 1)$ and characterizes the decay of the spectrum.*

*Suppose we seek the leading eigenpair $(\lambda_1, \mathbf{u}_1)$ starting from an initial vector $\mathbf{v}_0 \in \mathbb{R}^n$ with $\langle \mathbf{v}_0, \mathbf{u}_1 \rangle \neq 0$. Then, for any $q$, the Lanczos method converges to $\mathbf{u}_1$ with a strictly smaller error bound than power iteration under the same number of iterations.*

*Proof.* We analyze the convergence properties of the power iteration and Lanczos algorithms by estimating the error as the angle between the vector produced by each method and the true leading eigenvector.

**Step 1. Power iteration convergence.** Let $\mathbf{u}_1$ be the true leading eigenvector of $\mathbf{A}$ corresponding to $\lambda_1$, and let $\mathbf{v}_0$ be the initial vector before any iterations. After $k$ steps of power iteration, the normalized iterate is

$$\mathbf{v}_k = \frac{\mathbf{A}^k \mathbf{v}_0}{\|\mathbf{A}^k \mathbf{v}_0\|}. \tag{7}$$

**Lemma 3.2.** *Express $\mathbf{v}_0$ in the eigenbasis of $\mathbf{A}$ as*

$$\mathbf{v}_0 = \alpha_1 \mathbf{u}_1 + \alpha_2 \mathbf{u}_2 + \cdots + \alpha_n \mathbf{u}_n, \qquad \alpha_1 \neq 0. \tag{8}$$

*Then,*

$$\mathbf{A}^k \mathbf{v}_0 = \alpha_1 \lambda_1^k \mathbf{u}_1 + \alpha_2 \lambda_2^k \mathbf{u}_2 + \cdots + \alpha_n \lambda_n^k \mathbf{u}_n. \tag{9}$$

After normalization, the relative weight of $\mathbf{u}_i$ decays as $q^k$, where $q = \lambda_2/\lambda_1 < 1$. It follows from Golub & Van Loan (2013); Trefethen & Bau (1997) that the error satisfies

$$\sin \angle(\mathbf{v}_k, \mathbf{u}_1) \leq C(\mathbf{v}_0) q^k, \quad g(q) = q. \tag{10}$$

Thus, power iteration corresponds to applying the polynomial filter $p_k(x) = x^k$, which separates $\lambda_1$ from the rest of the spectrum poorly when the spectral decay is small.

**Step 2. Lanczos convergence.** The Lanczos method constructs approximations in the Krylov subspace

$$\mathcal{K}_m(\mathbf{A}, \mathbf{v}_0) = \mathrm{span}\{\mathbf{v}_0, \mathbf{A}\mathbf{v}_0, \ldots, \mathbf{A}^{m-1}\mathbf{v}_0\}, \tag{11}$$

which, at each step, introduces a new polynomial term of $\mathbf{A}$ applied to $\mathbf{v}_0$. The Ritz vectors are defined as

$$\mathbf{w}_i = \mathbf{Q}_m y_i, \tag{12}$$

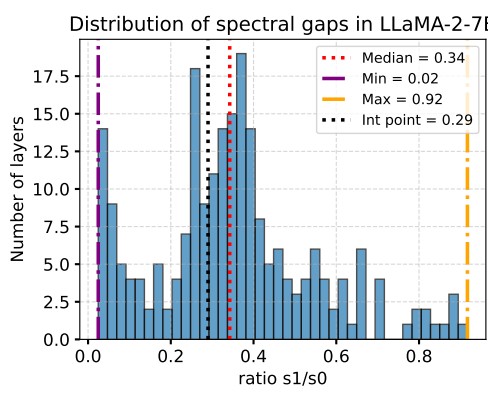 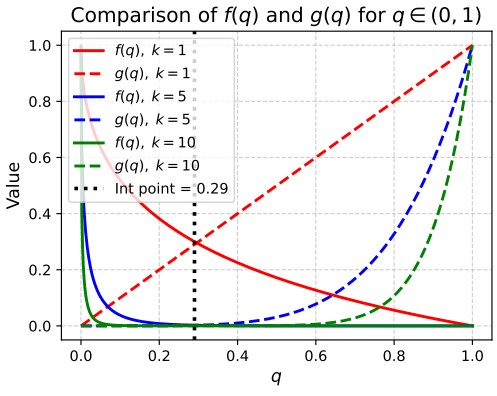

(a) Distribution of spectral gaps.  (b) Comparison of $f(q)$ and $g(q)$.

Figure 1: (a) Histogram of spectral gaps across layers in LLaMA-2-7B. In more than half of the layers, the spectral gap exceeds $0.29$, and on these layers we observe that $f(q) < g(q)$. (b) Curves $f^k(q)$ and $g^k(q)$ for $k \in \{1, 5, 10\}$. As $k$ increases, the curve $f^k(x)$ becomes tightly compressed against the horizontal axis, causing the area under it to shrink rapidly.

where $\mathbf{Q}_m$ is an orthonormal basis of $\mathcal{K}_m(\mathbf{A}, \mathbf{v}_0)$ and $y_i$ is an eigenvector of the projected tridiagonal matrix $\mathbf{T}_m = \mathbf{Q}_m^\top \mathbf{A} \mathbf{Q}_m$. These Ritz vectors lie in the Krylov subspace and approximate the true eigenvectors $\mathbf{u}_i$ of $\mathbf{A}$.

**Lemma 3.3.** *Let $\mathbf{w}_m$ denote the Ritz vector obtained after $m$ steps of the Lanczos algorithm, i.e., the approximation to $\mathbf{u}_1$ extracted from $\mathcal{K}_m(\mathbf{A}, \mathbf{v}_0)$. Then the error satisfies*

$$\sin \angle(\mathbf{w}_m, \mathbf{u}_1) \le 2 \left( \frac{1 - \sqrt{q}}{1 + \sqrt{q}} \right)^m, \quad f(q) = \frac{1 - \sqrt{q}}{1 + \sqrt{q}}, \tag{13}$$

*see, e.g., Parlett (1980); Saad (2003).*

**Step 3. Norm comparison.** We now compare the convergence rates of the two methods by analyzing the $L_2$ norms of their respective error functions over $q \in (0, 1)$:

$$I_k = \|f^k\|_{L_2(0,1)}^2 = \int_0^1 \left( \frac{1 - \sqrt{q}}{1 + \sqrt{q}} \right)^{2k} dq, \tag{14}$$

$$J_k = \|g^k\|_{L_2(0,1)}^2 = \int_0^1 q^{2k} \, dq = \frac{1}{2k + 1}. \tag{15}$$

With the substitutions $q = t^2$ and $u = \frac{1-t}{1+t}$, we obtain

$$I_k = 4 \int_0^1 u^{2k} \frac{1 - u}{(1 + u)^3} \, du. \tag{16}$$

Since $\frac{1}{(1+u)^3} \le 1$ for $u \in (0, 1)$, it follows that

$$I_k \le \frac{4}{(2k + 1)(2k + 2)} = \frac{2}{k + 1} \underbrace{\frac{1}{2k + 1}}_{J_k}. \tag{17}$$

Thus,

$$\|f^k\|_{L_2(0,1)} \le \sqrt{\frac{2}{k + 1}} \|g^k\|_{L_2(0,1)}. \tag{18}$$

Hence,

$$\|f^k\|_{L_2} - \|g^k\|_{L_2} \le 0 \qquad \forall \, k \ge 1, \tag{19}$$

and the difference between the two norms increases as $k$ grows.

$\square$

We show that within the spectral gap regime, the Lanczos method exhibits faster norm convergence compared to power iteration (Figure 1(b)). However, spectral gap statistics from real LLMs (Figure 1(a)) indicate that most values lie to the right of the intersection point of $f(q)$ and $g(q)$. This implies that, for the majority of layers, Lanczos converges faster not only in the integral sense but also pointwise.

# 4 EXPERIMENTS

We evaluated our approach using the YAQA pipeline, which consists of two stages: (1) estimation of Kronecker factors of the full-layer Hessian, and (2) a rounding-based quantization algorithm that leverages these factors.

Following the original YAQA work, for the second stage we adopted the QTIP quantization algorithm without fine-tuning, using the *quantlut_sym* decode mode with a bitshift codebook and the hyperparameters provided in the official repository [1].

For the baseline, we employed the **Sketch A** decomposition algorithm from the YAQA repository. In our method, we collected gradients on several minibatches of the calibration dataset and used the implementation of Algorithm 1 (**FastKron**) from the GFWSVD repository [2] to obtain the Kronecker factors.

We evaluated perplexity and zero-shot performance in downstream tasks for the LLaMA-2-7B, LLaMA-3-8B (Instruct), and Qwen-3-8B (Instruct) models. For the LLaMA models, we used a sequence length of 4096, and for Qwen — 2048. Runtime was measured using Python's built-in profiler, while the total number of tokens was computed as the product of the number of calibration sequences and their context length.

We also carried out QTIP experiment with identity Kronecker factors (reported as **No Hess** in the results), to demonstrate the advantage of incorporating second-order information for compression.

Table 1: Zero-shot accuracy for YAQA quantization of **LLaMA-2-7B**, comparing factors derived via power iteration (Sketch A) and FastKron. Lower is better for used resoures (↓), higher is better for accuracy (↑). "M" denotes millions and "K" denotes thousands of tokens.

| Method | Steps | Arc_c ↑ | Boolq ↑ | Piqa ↑ | Arc_e ↑ | HSwag ↑ | AVG ↑ | GPU/h ↓ | Tokens ↓ |
|---|---|---|---|---|---|---|---|---|---|
| 16 bit | – | **0.4325** | **0.7767** | **0.7774** | **0.7617** | **0.5721** | **0.6640** | – | – |
| 4 bit Sketch A | 4096 | 0.4274 | 0.7688 | 0.7752 | **0.7613** | **0.5672** | 0.6599 | 50 | 16 M |
| 4 bit FastKron | 75 | 0.4283 | 0.7792 | **0.7802** | 0.7610 | 0.5660 | 0.6629 | 5 | 712 K |
| 4-bit No Hess | – | **0.4352** | **0.7875** | 0.7742 | 0.7609 | 0.5628 | **0.6641** | – | – |
| 2 bit Sketch A | 4096 | 0.3805 | 0.7333 | 0.7562 | **0.7192** | **0.5227** | 0.6223 | 50 | 16 M |
| 2 bit FastKron | 150 | **0.3843** | **0.7510** | **0.7600** | 0.7112 | 0.5139 | **0.6240** | 6 | 1400 K |
| 2-bit No Hess | – | 0.2210 | 0.6355 | 0.6306 | 0.5152 | 0.3422 | 0.4689 | – | – |

Table 2: Zero-shot accuracy for YAQA quantization of **LLaMA-3-8B**, comparing factors derived via power iteration (Sketch A) and FastKron. Lower is better for used resoures (↓), higher is better for accuracy (↑).

| Method | Steps | Arc_c ↑ | Boolq ↑ | Piqa ↑ | Arc_e ↑ | HSwag ↑ | AVG ↑ | GPU/h ↓ | Tokens ↓ |
|---|---|---|---|---|---|---|---|---|---|
| 16 bit | - | **0.5171** | **0.8409** | **0.7986** | **0.8177** | **0.5908** | **0.7131** | – | — |
| 4-bit Sketch A | 4096 | **0.5136** | **0.8443** | 0.7997 | 0.8198 | **0.5865** | 0.7127 | 92 | 16 M |
| 4-bit FastKron | 75 | 0.5116 | 0.8438 | **0.8025** | **0.8207** | 0.5863 | **0.7129** | 9.5 | 712 K |
| 4-bit No Hess | – | 0.5119 | 0.8415 | 0.7959 | 0.8097 | 0.5859 | 0.7112 | – | – |
| 2-bit Sketch A | 4096 | **0.4312** | 0.7567 | 0.7647 | 0.7391 | **0.5259** | 0.6435 | 92 | 16 M |
| 2-bit FastKron | 100 | 0.4277 | **0.7646** | **0.7661** | **0.7468** | 0.5159 | **0.6442** | 11.5 | 950 K |
| 2-bit No Hess | – | 0.2363 | 0.6336 | 0.6554 | 0.5108 | 0.3620 | 0.5094 | – | – |

---

[1] https://github.com/Cornell-RelaxML/qtip
[2] https://github.com/sayankotor/FisherKronecker/

Table 3: Zero-shot accuracy for YAQA quantization of **Qwen3-8B**, comparing factors derived power iteration (Sketch A) and FastKron. Lower is better for used resourses ($\downarrow$), higher is better for accuracy ($\uparrow$).

| Method | Steps | Arc_c $\uparrow$ | Boolq $\uparrow$ | Piqa $\uparrow$ | Arc_e $\uparrow$ | HSwag $\uparrow$ | AVG $\uparrow$ | GPU/h $\downarrow$ | Tokens $\downarrow$ |
|---|---|---|---|---|---|---|---|---|---|
| 16 bit | - | **0.5563** | **0.8682** | **0.7677** | **0.8354** | **0.5708** | **0.7197** | — | — |
| 4-bit Sketch A | 4096 | **0.5503** | 0.8611 | 0.7612 | 0.8324 | 0.5601 | **0.7132** | 84 | 8 M |
| 4-bit FastKron | 150 | 0.5469 | 0.8667 | 0.7601 | **0.8287** | **0.5637** | **0.7132** | 42 | 712 K |
| 4-bit No Hess | – | 0.5467 | **0.8675** | **0.7622** | 0.8312 | 0.5585 | **0.7132** | – | – |
| 2-bit Sketch A | 4096 | 0.4536 | 0.7782 | **0.7435** | **0.7797** | 0.4611 | 0.6432 | 84 | 8 M |
| 2-bit FastKron | 150 | **0.4616** | 0.8416 | 0.7334 | 0.7702 | **0.4853** | **0.6584** | 42 | 712 K |
| 2-bit No Hess | – | 0.3993 | **0.8675** | 0.7743 | 0.7003 | 0.4758 | 0.6434 | – | – |

## 5 RESULTS

The results for zero-shot downstream tasks are in the Tables 1, 2, 3. The results show that accuracy degradation under 4-bit quantization is negligible: for LLaMA-2 7B, the task average decreases by 0.41% when factors are estimated via Sketch A, and by only 0.11% with FastKron. For 2-bit quantization, the average drop is around 4%.

For the instruction-tuned LLaMA-3 8B model, 4-bit quantization leads to a 0.05% drop with Sketch A and 0.02% with FastKron, while 2-bit quantization is substantially more challenging, yielding drops of about 9.7% in both cases.

In both settings, FastKron achieves comparable or better accuracy while converging with roughly 10× lower compute budget (e.g., 5 vs. 50 GPU hours, or 9–11 vs. 90 GPU hours). Moreover, it consistently yields several tens of percent higher accuracy compared to Sketch A. Calibration with FastKron requires only ≈1M tokens (the exact number depends on the number of microbatches), while Sketch A requires 16M.

A comparison with the QTIP-based method without second-order information (No Hess) shows that, for 4-bit quantization, the absence of second-order terms has little impact — and can even lead to slightly better results. However, with 2-bit precision, they become critical, leading to a 12–13% reduction in accuracy compared to factor-based approaches. This result further demonstrates that second-order information becomes increasingly important under more aggressive quantization regimes, making efficient factor computation highly relevant in such settings.

For Qwen 3, experiments were conducted at half the sequence length, resulting in a smaller speedup of about 2× at comparable budgets. The average downstream accuracy remains nearly unchanged at 4-bit, while at 2-bit the drop is about 6% and 5%, respectively.

Perplexity results (Table 4) follow the same trend as zero-shot accuracy, with the exception that FastKron-based methods yield several percent lower perplexity than Sketch A.

The corresponding code will be provided in the supplementary material, and links to the released checkpoints will be included in the final version.

## 6 CALIBRATION TOKEN COUNT AND PERFORMANCE

Table 4: Perplexity (lower $\downarrow$ is better) on WikiText and C4 for three models.

| Method | LLaMA-2-7B | | LLaMA-3-8B | | Qwen3-8B | |
|---|---|---|---|---|---|---|
| | Wiki $\downarrow$ | C4 $\downarrow$ | Wiki $\downarrow$ | C4 $\downarrow$ | Wiki $\downarrow$ | C4 $\downarrow$ |
| 16 bit | **5.11** | **6.63** | **6.00** | **8.40** | **8.99** | **12.48** |
| 4-bit Sketch A | **5.17** | **6.69** | **6.88** | **9.96** | 9.29 | 12.72 |
| 4-bit FastKron | 5.18 | 6.71 | 6.89 | 10.02 | **9.16** | **12.66** |
| 2-bit Sketch A | **6.18** | **8.00** | **8.98** | **12.79** | 16.04 | 18.21 |
| 2-bit FastKron | 6.40 | 8.31 | 9.11 | 12.98 | **13.35** | **16.86** |

In the previous section, we showed that FastKron achieves the same downstream quality with a substantially smaller budget. Increasing the number of calibration tokens results in a larger number of microbatches, thereby reducing the variance of the estimates and yielding more stable Kronecker factors for the compression method. This naturally raises the question: How sensitive is FastKron's downstream performance to the number of tokens used during calibration?

To isolate the effect of the calibration token budget, we varied the number of gradient-collection steps between 75 and 200, while keeping all other components of the FastKron pipeline fixed. For each configuration, Kronecker factors were computed from the collected calibration data and then used in the quantization pipeline.

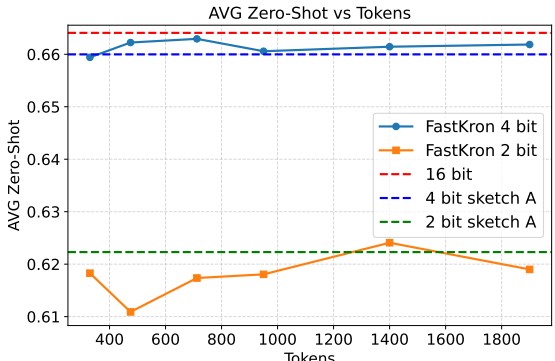

Figure 2: Average zero-shot validation performance as a function of calibration token count for 4-bit and 2-bit quantization of the LLaMA-2 7B model. The red dashed line shows the average accuracy of the uncompressed model, while the blue and green lines correspond to Sketch A computed on 16M tokens.

The complete results for the perplexity and zero-shot validation are provided in Appendix A, and Figure 2 shows the average zero-shot validation performance as a function of the number of tokens.

We expected downstream performance to improve with a larger number of microbatches, but Figure 2 shows that for 4-bit quantization, the performance remains nearly constant, with a small local maximum at 712K tokens (75 steps). For 2-bit quantization, the performance is more variable, peaking at 1.4M tokens (150 steps), but the difference never exceeds 1.5%. This stability further underscores the practicality of FastKron for large-scale deployment.

## 7 CONCLUSION

We investigated second-order post-training quantization for large language models and proposed FastKron, a practical replacement for the Kronecker-factor estimation step in YAQA.

Our empirical evaluation on LLaMA-2-7B, LLaMA-3-8B (Instruct), and Qwen-3-8B (Instruct) demonstrates that incorporating second-order information consistently improves PTQ robustness. FastKron achieves the same downstream accuracy as the power iteration baseline — nearly the same quality at 4-bit compression and a 5–6% drop under 2-bit quantization — while reducing factor-computation time by up to $10\times$ and requiring orders of magnitude fewer calibration tokens. The experiments also show that incorporating second-order information becomes crucial under more extreme quantization: with factor-aware methods, we observe similar quality at 4-bit precision and up to 12–13% higher accuracy at 2-bit precision.

Overall, FastKron is a drop-in, second-order Kronecker factor estimator that makes curvature-aware PTQ feasible at LLM scale. It maintains quantization accuracy, reduces computation and token requirements, and ensures stable performance across reasonable calibration budgets. By bridging the gap between theoretical efficiency and practical scalability, FastKron brings second-order PTQ closer to becoming a deployable tool for compression and inference of billion-parameter language models.

### ETHICS STATEMENT

This work focuses on methods for improving the efficiency and practicality of post-training quantization of large language models. Our research does not involve human subjects, personally identifiable information, or sensitive data. All experiments are conducted on publicly available models (LLaMA, Qwen) and benchmarks (ARC, BoolQ, PIQA, HellaSwag, WikiText, C4), ensuring reproducibility and transparency. We do not release any new datasets containing personal or private

data. The proposed methods are intended for reducing the computational cost and energy consumption of deploying large models, which we view as a positive contribution to sustainability. We are not aware of any direct negative societal impacts, though—as with any model compression technique—improved efficiency could indirectly facilitate the deployment of large models in settings where misuse is possible. We encourage responsible use of these methods in accordance with the ICLR Code of Ethics.

## REPRODUCIBILITY STATEMENT

We have made every effort to ensure that our results are fully reproducible. The implementation of the proposed method is provided in the supplementary materials. All theoretical assumptions and proofs are included in the main text. Links to the baseline repositories required to reproduce our experiments are provided in Section 4. Additionally, all model checkpoints used in our experiments will be released publicly in the camera-ready version of the paper.

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

## A    APPENDIX A

We report extended quantization results for Token Count ablation in Tables 5, 6.

Table 5: Perplexity and zero-shot accuracy for 4-bit YAQA quantization of LLaMA-2-7B, comparing factors derived via power iteration (Sketch A) and GFWSVD. Lower is better for Perplexity ($\downarrow$), higher is better for accuracy ($\uparrow$).

| Method | Wiki$\downarrow$ | C4$\downarrow$ | Arc_c$\uparrow$ | Boolq$\uparrow$ | Piqa$\uparrow$ | Arc_e$\uparrow$ | HSwag$\uparrow$ | AVG$\uparrow$ | Steps | Tokens |
|---|---|---|---|---|---|---|---|---|---|---|
| 16 bit | **5.11** | **6.63** | **0.4325** | 0.7767 | 0.7774 | **0.7617** | **0.5721** | **0.6640** | – | – |
| 4 bit Sketch A | **5.17** | **6.69** | 0.4274 | 0.7688 | 0.7752 | 0.7613 | 0.5672 | 0.6599 | – | 16M |
| 4 bit FastKron | 5.19 | 6.71 | 0.4241 | 0.7697 | 0.7780 | 0.7579 | 0.5674 | 0.6594 | 35 | 330K |
| 4 bit FastKron | 5.19 | 6.71 | **0.4291** | 0.7764 | 0.7780 | 0.7601 | **0.5676** | 0.6622 | 50 | 475K |
| 4 bit FastKron | 5.18 | 6.71 | 0.4283 | **0.7792** | **0.7802** | 0.7610 | 0.5660 | **0.6629** | 70 | 712K |
| 4 bit FastKron | 5.19 | 6.72 | 0.4197 | 0.7776 | 0.7780 | 0.7615 | 0.5661 | 0.6605 | 100 | 950K |
| 4 bit FastKron | 5.18 | 6.72 | 0.4257 | 0.7737 | 0.7786 | **0.7622** | 0.5670 | 0.6614 | 150 | 1400K |
| 4 bit FastKron | 5.18 | 6.71 | 0.4266 | 0.7776 | 0.7780 | 0.7605 | 0.5666 | 0.6618 | 200 | 1900K |

## B    APPENDIX B: LLM USAGE STATEMENT

We used large language models (LLMs) only as a general-purpose writing assistant for grammar checking and text polishing. The research ideas, implementation, analysis, and conclusions are entirely our own.

Table 6: Perplexity and zero-shot accuracy for 2-bit YAQA quantization of LLaMA-2-7B, comparing factors derived via power iteration (Sketch A) and GFWSVD. Lower is better for Perplexity ($\downarrow$), higher is better for accuracy ($\uparrow$).

| Method | Wiki$\downarrow$ | C4$\downarrow$ | Arc_c$\uparrow$ | Boolq$\uparrow$ | Piqa$\uparrow$ | Arc_e$\uparrow$ | HSwag$\uparrow$ | AVG$\uparrow$ | Steps | Tokens |
|---|---|---|---|---|---|---|---|---|---|---|
| 16 bit | **5.11** | **6.63** | **0.4325** | **0.7767** | **0.7774** | **0.7617** | **0.5721** | **0.6640** | – | – |
| 2 bit Sketch A | 6.18 | 8.00 | 0.3805 | 0.7333 | 0.7562 | 0.7192 | 0.5227 | 0.6223 | – | 16M |
| 2 bit FastKron | 6.59 | 8.49 | 0.3899 | 0.7232 | 0.7573 | 0.7034 | 0.5176 | 0.6182 | 35 | 330K |
| 2 bit FastKron | 6.44 | 8.40 | 0.3658 | 0.7152 | 0.7568 | 0.7032 | 0.5135 | 0.6109 | 50 | 475K |
| 2 bit FastKron | 6.44 | 8.30 | 0.3720 | 0.7320 | 0.7579 | 0.7112 | 0.5137 | 0.6173 | 70 | 712K |
| 2 bit FastKron | **6.43** | 8.36 | 0.3677 | 0.7393 | 0.7578 | 0.7128 | 0.5127 | 0.6180 | 100 | 950K |
| 2 bit FastKron | 6.40 | **8.31** | **0.3843** | **0.7510** | **0.7600** | 0.7112 | **0.5139** | **0.6240** | 150 | 1400K |
| 2 bit FastKron | 6.47 | 8.39 | 0.3618 | 0.7486 | 0.7540 | 0.7115 | 0.5091 | 0.6190 | 200 | 1900K |

