# OpenReview forum: "Fast and Accurate Fisher-Guided Quantization via Efficient Kronecker Factor Approximation"
_ICLR.cc/2026/Conference — ICLR 2026 Conference Withdrawn Submission_

### Official Review · Reviewer_AkSJ · 2025-10-21

**Soundness:** 2
**Presentation:** 2
**Contribution:** 2
**Rating:** 2
**Confidence:** 4

**Summary:**

This paper proposes a different way of estimating the Hessian for the YAQA rounding algorithm. The proposed method is cheaper to compute and slightly worse than YAQA's Sketch A, and worse than YAQA's Sketch B.

**Strengths:**

- The proposed Hessian sketch is much cheaper than both of YAQA's Hessian sketches.
- There is some nice analysis on the convergence rate of power iteration and the Hessian sketch used.

**Weaknesses:**

- This paper seems somewhat incomplete and hastily written. There are relatively few empirical results, the Appendix is almost nonexistent (although this is not an issue on its own, it is odd that a full conference submission would have almost no Appendix), and some of the names in the bibliography are literally wrong.
- The main contribution of the paper is taking the Hessian sketch from GFWSVD (Chekalina et al. (2025)) and applying it in the YAQA framework. This sketch directly estimates the Fisher, whereas Sketch A (the worse of the two YAQA sketches and the main baseline here) is actually a biased estimate of the Fisher. Sketch B would be a conceptually closer baseline to the one presented here, and Sketch B performs much better than this.

**Questions:**

- The paper says that it introduces FastKron, but L287 implies that FastKron was introduced in an earlier code. Furthermore, the authors link to a public repo for the GFWSVD paper. Is FastKron new or not?
- My score is partly based off the assumption that FastKron is from GFWSVD, which implies that there isn't actually much going on in this paper. It combines GFWSVD and YAQA, both of which are existing works in the literature. Perhaps the authors can clarify this detail.

---

### Official Review · Reviewer_bCLe · 2025-10-21

**Soundness:** 2
**Presentation:** 3
**Contribution:** 1
**Rating:** 2
**Confidence:** 4

**Summary:**

This work modifies and improves the YAQA algorithm of Tseng et al (2025b), introducing a method called FastKron which uses Lanczos iteration in place of plain power iteration to estimate Kronecker factors of a structured approximation to the Hessian of loss w.r.t. weights. FastKron is shown to have faster convergence than power iteration when the largest two eigenvalues are close. Practical results for YAQA quantisation of pretrained LLMs using QTIP show FastKron achieving similar downstream accuracy to "Sketch A" from the YAQA paper, with fewer tokens and much shorter wall clock time.

---
_Tseng et al (2025b): Model-Preserving Adaptive Rounding (YAQA)_

**Strengths:**

The technique is well-motivated, and the core idea is sound. In particular, it is sensible to look for a scheme with faster convergence than power iteration, saving PTQ compression time. The wall clock speedups presented are substantial.

Equations, tables and plots are generally clear, and the paper includes all of the structural elements required to back up the author's claims. The work makes the contribution clear, outlining the provenance of ideas from Tseng et al. (2025b) and Chekalina et al. (2025), which are combined in FastKron.

---
_Chekalina et al. (2025): Generalized Fisher-Weighted SVD: Scalable Kronecker-Factored Fisher Approximation for Compressing Large Language Models (GFWSVD)_

**Weaknesses:**

My main concern with this work is that it is an insufficient step on top of the YAQA and GFWSVD component techniques to be of interest to the community as a full conference paper. Specifically, neither the method nor results convey enough new information for me to recommend acceptance. If the method of FastKron were supplemented by a substantial theoretical insight and excellent empirical investigation, this challenge of incrementalism could be overcome, but the theory and experiments do not meet this bar.

Specific concerns:
 - The method is not clearly and comprehensively explained. The reader must piece together Eqn. 4, as a fixed-point iteration equation, with Algorithm 1, where "leading singular triplet" line incorporates the Lanczos algorithm, and relies on the fact that $\tilde{I}_F \mathbf{z}$ can be computed efficiently using the method described in Chekalina et al. (2025), Section 4.1.
 - Algorithm 1 is functionally identical to Chekalina et al., Algorithm 1, so does not help demonstrate the unique contribution of FastKron over GFWSVD.
 - Theorem 3.1 is not demonstrated as stated, at least by Step 3. I think where it says "for any q ... strictly smaller error" should read e.g. "for a uniform distribution of $q \in [0, 1]$ ... smaller expected error", unless I misunderstand the result.
   - If this is so, I question the relevance of this theorem - there is no reason given to suspect a uniform distribution of spectral gap, and Figure 1a indicates the spectral gap is non-uniform in practice.
 - At least one result (LLaMA-2 7B, AVG Zero-Shot, 2-bit) appears to use an overfitted choice of calibration token count. In Figure 2, the performance varies considerably, and 1400K tokens outperforming ~1900K tokens. In Table 1, the 1400K token result is reported with a performance consistent with the Figure 2 peak, and highlighted in bold as outperforming the Sketch A baseline. It is hard to reach the same conclusion regarding FastKron vs Sketch A from inspecting the table and figure, highlighting the poor experimental practice. If selecting token count from a search, this search should be performed on a validation set, or based on the intrinsic YAQA objective, or reduced into a simple procedure for choosing the number of steps, which can be consistently applied across models.
 - It isn't clear why YAQA Sketch B is not included in the results, as this seems to be a relevant baseline.
 - Tseng et al. (2025b) claim in Section 3.2.1 that Sketch A takes around 20 GPU-hours for a 10B model / 20M tokens, where this work measures 50 or 92 hours (Tables 1, 2), with no explanation offered for the difference.

Minor concerns:
 - Eqn. 6 seems to miss $\left<\right>$ when compared with Tseng et al. (2025b), Eqn 8, 9.
 - To help the reader follow the derivation of Theorem 3.1, the paper would benefit from an algorithmic view of the Lanczos method to supplement the high-level explanation of L211 - showing the concrete steps involved.
 - $m$ steps of Lanczos iteration is introduced in Eqn 11, presumably no relation to $m$ of L043, then substituted with $k$ steps in Figure 1b and Eqn 14. This is confusing --- if it's functionally identical to $k$, this should be used consistently.
 - Definitions of these symbols: unclear $\mathcal{R}$ (Algorithm 1), unnecessary/unclear: $g$, $p_k$, $f$.

**Questions:**

I would appreciate any clarifications or responses to the points listed above as "main concern" and "specific concerns".

---

### Official Review · Reviewer_bWh3 · 2025-11-01

**Soundness:** 3
**Presentation:** 2
**Contribution:** 2
**Rating:** 4
**Confidence:** 4

**Summary:**

The authors proposes replacing the power iteration method in YAQA with FastKron (renamed from GFWSVD), a more efficient Kronecker decomposition method. Experiments on a wide range of models suggest that their method is $10\times$ faster in obtaining the Kronecker factors while achieving the same downstream accuracy.

**Strengths:**

1. FastKron speeds up Kronecker-factor estimation by $10\times$ while achieving the same downstream accuracy.
1. This is a bridge paper that connects the modern LLM quantization algorithms with numerical linear algebra.
1. Section 3 provides theoretical justification for why FastKron (Lanczos-base) is faster than the power iteration used in YAQA: it's because the spectral gaps of real LLM layers fall into the regime where Lanczos-based methods perform better.
1. Section 6 explores the relationship between Kron’s downstream performance to the number of tokens used during calibration.

**Weaknesses:**

1. FastKron/GFWSVD was already proposed in [another paper](https://arxiv.org/abs/2505.17974), which significantly undermines the novelty of this paper.
1. Speeding up the quantization process (as opposed to dequantization/inference/decoding) has limited impact, because each model only needs to be quantized once up front.
1. Insufficient baselines: the authors didn't compare FastKron against other Fisher Kronecker factor estimators like K-FAC, EKFAC, FWSVD, and TFWSVD.
1. Insufficient experiments: all experiments are ran on 8B models, but ideally we want to see the performance across different scales.

**Questions:**

1. In Tables 1 through 3, did you mean to say "GPU*h", i.e., GPU hours, instead of "GPU/h"?
1. In Tables 1 though 3, when/why does "No Hess" occasionally perform as good as Hessian-based quantizers?
1. In Table 3, why does FastKron only get a $2\times$ speedup on Qwen3-8B? Is it because it has a difference spectral gap structure, or context length difference, or is there anything special about its architecture?

---

### Official Review · Reviewer_KBG6 · 2025-11-01

**Soundness:** 2
**Presentation:** 3
**Contribution:** 2
**Rating:** 4
**Confidence:** 4

**Summary:**

The paper proposes to improve the computational cost of the YAQA quantization framework by replacing the power iteration method with FastKron to compute the Kronecker product approximation of the Hessian. Theoretically, they show that the Lanczos method converges faster than the power iteration method. They experimentally evaluate their method on LLaMA-2 7B, LLaMA-3 8B Instruct, Qwen 3 8B Instruct to show the benefits.

**Strengths:**

1. They propose a novel method to improve the computational cost of YAQA method
2. The convergence rates of the power iteration method and Lanczos method are analyzed theoritically
3. Their method is evaluated emperically on three models on downstreams tasks and perplexity on C4 and WikiText.

**Weaknesses:**

1. The Algorithm 1 and the theoretical analysis seem heavily borrowed from prior work.
2. No comparision to other baseline methods
3. Their method performs worse on the perplexity of Llama 2 class of models

**Questions:**

1. How does the FastKron-based method scale for even larger models? How does it scale for smaller models?
2. Is the method limited only to Language models? How does it work for Diffusion or Flow models for example
3. Does the calibration dataset or its size matter?

---

### Note · Authors · 2026-01-04

I have read and agree with the venue's withdrawal policy on behalf of myself and my co-authors.